# The PKD-Dependent Biogenesis of TGN-to-Plasma Membrane Transport Carriers

**DOI:** 10.3390/cells10071618

**Published:** 2021-06-28

**Authors:** Yuichi Wakana, Felix Campelo

**Affiliations:** 1School of Life Sciences, Tokyo University of Pharmacy and Life Sciences, Hachioji, Tokyo 192-0392, Japan; 2ICFO-Institut de Ciencies Fotoniques, The Barcelona Institute of Science and Technology, 08860 Barcelona, Spain

**Keywords:** CARTS, Golgi complex, membrane contact sites, membrane trafficking, protein kinase D, protein sorting, lipid metabolism

## Abstract

Membrane trafficking is essential for processing and transport of proteins and lipids and to establish cell compartmentation and tissue organization. Cells respond to their needs and control the quantity and quality of protein secretion accordingly. In this review, we focus on a particular membrane trafficking route from the *trans*-Golgi network (TGN) to the cell surface: protein kinase D (PKD)-dependent pathway for constitutive secretion mediated by carriers of the TGN to the cell surface (CARTS). Recent findings highlight the importance of lipid signaling by organelle membrane contact sites (MCSs) in this pathway. Finally, we discuss our current understanding of multiple signaling pathways for membrane trafficking regulation mediated by PKD, G protein-coupled receptors (GPCRs), growth factors, metabolites, and mechanosensors.

## 1. Introduction

Cells synthesize complex biological molecules—nucleic acids, carbohydrates, proteins, and lipids—to support life. Importantly, cells can sustain life not only by synthesizing these molecules, but also by precisely organizing them in space and time [1]. In particular, eukaryotic cells have evolved a complex intracellular organization, which is facilitated by the existence of membrane-bound organelles: functional entities with a well-defined biochemical composition. Because of this compartmentalization of biochemical reactions, eukaryotic cells can efficiently and simultaneously perform a large number of biological functions. How these different functions are coordinated in a holistic fashion is one of the key questions in cell physiology.

Membrane trafficking plays a crucial role in processing and transport of proteins and lipids, thus underlying the maintenance of organelle composition and identity. In addition, the correct delivery of cargo proteins to the cell surface or their release to the extracellular space is fundamental to cell-to-cell communication, which influences virtually all cellular functions. Included in these cargo proteins are the cell surface receptors such as integrins (required for cell adhesion and migration), cytokines and growth factors (with roles in cell proliferation, differentiation, and wound healing), extracellular matrix (ECM) proteins such as collagens and matrix metalloproteinases (needed for the assembly of tissues in multicellular organisms), neurotransmitters (necessary for neural functions), or mucins (needed to create protective barriers in epithelial tissues) [2]. These examples illustrate the global significance of membrane trafficking and protein secretion in cell physiology, especially highlighting their essential roles in cell growth and development, which require expansion and remodeling of cellular membranes. Importantly, although most cellular functions are interdependent on one another, how the secretory pathway is regulated in response to physiological needs still remains poorly understood [3,4,5]. In this review, we highlight our current knowledge on how cells sense internal and external signals, relay them, and integrate them to adapt the secretory machinery. In particular, we focus here on reviewing how multiple signaling pathways converge to control protein export from the Golgi apparatus.

### 1.1. Membrane Trafficking

Membrane trafficking along the secretory pathway involves the movement (trafficking) of biochemical molecules (mainly proteins and lipids) by means of membrane-bound transport intermediates. These transport intermediates shuttle biomolecules from one membrane-bound organelle to another one or for delivery to the plasma membrane (PM) or secretion to the extracellular space [6]. The different organelles of the secretory pathway include the endoplasmic reticulum (ER), the Golgi apparatus, and the endo-lysosomal system. Newly synthesized proteins that contain a signal sequence or signal peptide—a short amino-terminal stretch of hydrophobic amino acids—are targeted to the secretory pathway by co-translational translocation into the ER [7]. After proteolytic cleavage of the signal sequence, soluble proteins end up in the ER lumen whereas transmembrane proteins remain inserted in the ER membrane [7]. In the ER, proteins acquire initial glycosylations and are correctly folded, after which they either remain in the ER as resident proteins or are exported from the ER by COPII-dependent transport carriers directed towards the ER-Golgi intermediate compartment (ERGIC) and/or the Golgi apparatus (from now on, simply the Golgi) [8].

In mammalian cells at interphase, the Golgi is composed of multiple stacks of 4–12 flattened cisternae [9]. Usually, the different Golgi stacks are laterally linked, forming the so-called Golgi ribbon. The individual Golgi stacks are transversely polarized following a *cis*/medial/*trans* organization [10]. The *cis*-cisternae receive cargoes from the ER/ERGIC [11]. On the opposite side, the *trans*-Golgi network (TGN) serves as the exit station for anterograde transport out of the Golgi. After entering at the *cis*-face of the Golgi, some cargo proteins (such as some cargo receptors and ER chaperones) are recycled back to the ER/ERGIC in COPI-coated transport carriers. The remaining cargo proteins follow their intracellular route across the Golgi.

Besides being polarized with respect to the directionality of cargo flow, the Golgi is also polarized in terms of its biochemical composition because Golgi-resident enzymes are not homogeneously allocated across the stack, but they rather have a polarized distribution. Specifically, early-acting glycosylation enzymes are enriched in the *cis*-cisternae, whereas late acting enzymes mainly localize to *trans*-cisternae. This polarized organization of Golgi enzymes guarantees the sequential glycosylation of cargo proteins while being trafficked within the Golgi [12]. Once fully processed cargoes reach the TGN, they are sorted for their export in specific transport carriers for various destinations within the cell.

### 1.2. Multiple Export Pathways from the TGN

Various mechanisms have been proposed to be at play for the retention of Golgi residents at their sites of action [12,13,14]. The sorting of secretory cargoes is thought to occur predominantly at the level of the TGN, which, accordingly, is sometimes called the central sorting station of the secretory pathway. At the TGN, cargo molecules are sorted for export into specific transport carriers with different characteristics and destinations. In particular, cargo-containing TGN-derived transport carriers can be directed to the endo-lysosomal system (early endosomes, late endosomes/lysosomes), the cell surface (basolateral or apical PM in polarized cells), secretory granules (especially in professional secretory cells), or even back to earlier Golgi cisternae. Notably, even cargoes that are destined to the same membrane can be sorted for export into distinct transport carriers. For instance, there are different export routes controlled by different molecular machinery for the export of cargoes to the PM [15]. Despite this specificity, we can list some key proteins commonly shared between multiple exit pathways from the TGN: small GTPases of the ARF and Rab family (needed to recruit diverse effectors to the TGN and transport carriers), adaptor protein complexes (with roles in sorting of cargoes and often binding to the clathrin coat), microtubule-associated kinesin motors (needed for the elongation of transport carrier precursors and for carrier transport through the cytosol). However, it is not the aim of this review to elaborate on molecular machinery in different export routes from the TGN. For that, we refer the readers to a number of excellent reviews on these topics [2,15,16,17,18,19,20]. Here, we will focus on reviewing our current understanding of a particular export route from the TGN to the PM: the protein kinase D (PKD)-dependent pathway for constitutive secretion mediated by carriers of the TGN to the cell surface (CARTS). We highlight the machinery and molecular mechanisms known to be involved in CARTS biogenesis and establish possible links and differences between PKD-dependent constitutive secretion routes.

## 2. PKD Is a Key Regulator of TGN Export and Golgi Lipid Homeostasis

### 2.1. Discovery of PKD as a Central Regulator of TGN Export

Nearly 30 years ago, work from the Malhotra lab showed that ilimaquinone, a metabolite obtained from the sea sponge *Hippospongia metachromia*, causes the extensive conversion of sac-like Golgi cisternae into small vesicular structures, a phenotype suggested to result from uncontrolled membrane fission [21]. These effects, which are reversible, parallel the inhibition of anterograde cargo export from the Golgi [21]. Subsequent investigations showed that ilimaquinone-induced Golgi vesiculation is independent of ARF1, Rabs and COPI proteins, but required ßγ subunits of heterotrimeric G proteins [22] and, downstream of that, the kinase activity of PKD [23]. PKD had been found to localize at the Golgi membranes and suggested to play a role in protein secretion [24].

The exogenous expression of a kinase-dead, dominant-negative mutant of PKD (PKD-KD) induces the formation of long membrane tubules emanating from, but continuous with, the TGN [25]. Notably, PKD-KD-induced TGN-derived tubules contain cargoes that are normally exported from the TGN, such as the transmembrane proteins TGN46 and furin [25]. Indeed, expression of PKD-KD or siRNA-mediated depletion of PKD inhibits the secretion of model cargoes, such as signal sequence horseradish peroxidase (ss-HRP) [25,26,27,28,29]. Collectively, these studies underscored the importance of PKD kinase activity for transport carrier fission and placed PKD as a central regulator of TGN-to-cell surface trafficking [30,31] (Table 1).

PKD is a family of serine/threonine-protein kinases that, in humans, encompasses 3 canonical members: PKD1 (also known as PKCµ; gene name *PRKD1*), PKD2 (gene name *PRKD2*), and PKD3 (also known as PKCν; gene name *PRKD3*) [31,58]. Although no PKD ortholog has been described in yeast, PKD family members are conserved throughout evolution from nematodes to humans [59]. PKD plays different functions in cell physiology [58], such as (i) promoting export from the TGN by assisting in the membrane fission reaction [17,25,26,30,31,58]; (ii) mediating cell migration by controlling integrin recycling and recruitment in focal adhesions [28,52]; (iii) contributing to cell survival under oxidative stress [60]; and (iv) activating inflammasomes to initiate the innate immune response [61]. Common to these functions is that PKD acts in response to metabolic flow of the lipid second-messenger diacylglycerol (DAG) [32,58,62,63].

In 2008, the Olson lab reported that PKD1 knockout mice are embryonically lethal, whereas mice with a cardiac-specific deletion of PKD1 are viable [64]. Consistently, the Cantrell lab showed that the homozygous expression in mice of a catalytically inactive PKD1 mutant allele is embryonically lethal [65]. By contrast, both catalytically inactive PKD2 knock-in mice and PKD2 knockout mice were viable and born at the expected Mendelian frequency [65]. However, the Harada lab reported that both PKD1 and PKD2 single knockout mice were viable and born at a normal frequency [66]. Although PKD1 and PKD2 double knockout mice are embryonically lethal, these authors reported that mouse embryonic fibroblasts (MEFs) deficient in all three PKD family members had no obvious defects in transport of anterograde transport markers such as vesicular stomatitis virus G protein (VSV-G) or glycosylphosphatidyl inositol (GPI)-anchored protein (AP) [66]. Nevertheless, matrix metalloproteinase (MMP) secretion was inhibited in MEFs from catalytically inactive PKD2 knock-in mice [29]. Finally, work from the Jin lab showed that PKD3 knockout mice are viable, but present morphological liver defects and exhibit age-dependent spontaneous hepatic fibrosis [67]. These results are thought to reflect the multiple functions of PKDs and their functional redundancy during development. In this review, we focus on the roles that PKD plays in protein export from the TGN (additional functions of this family of protein kinases have been exhaustively reviewed in [58,68]).

### 2.2. PKD Structure

PKD isoforms are modular proteins that consist of several domains with defined functions. At the N terminus, PKD1 and PKD2, but not PKD3, contain a short amino acid stretch rich in prolines and alanines, which is postulated to act as a membrane insertion domain [31,69]. PKD also contains a ubiquitin-like domain (ULD) that allows for homo- and hetero-dimerization [27,59,70,71]. Next, as part of its N-terminal regulatory domain, PKD contains a pair of cysteine-rich domains, termed C1a and C1b, respectively. The C1a is a DAG-binding domain [32,62], and so it is the C1b domain, although with a lower affinity [72]. The C1b has also been reported to mediate the interaction of PKD with the small G protein ARF1 [73]. Following those, PKD contains an auto-inhibitory pleckstrin homology (PH) domain [74], and finally, at its C terminus, the catalytic kinase domain, which encodes the enzymatic activity, as well as a postsynaptic density-95/discs large/zonula occludens-1 (PDZ)-binding domain in PKD1 and PKD2, responsible for substrate binding [75] (Figure 1).

PKD1 is a 912 amino acid long protein (molecular weight ~102 kDa), PKD2 is an 878 amino acid long protein (molecular weight ~97 kDa), and PKD3 is an 890 amino acid long protein (molecular weight ~100 kDa). The amino acid sequence of the three PKD members is highly conserved, with PKD1 and PKD2 sharing the highest similarity (~85% amino acid identity in mammals) [65]. Regarding its intracellular localization, different PKD members have been localized to different locations depending on the cell type and also as a response to different stimuli [65,76,77].

### 2.3. PKD Roles in Constitutive and Regulated Protein Secretion

PKD is involved in both regulated and constitutive secretion. On the one hand, in specialized secretory cells, PKD is required for the efficient formation of secretory granules and therefore for the regulated exocytosis of some cargoes, such as chromogranin A [35,36] and insulin [33] (Table 1). Interestingly, Ricci and colleagues showed that in nutrient-starved pancreatic ß cells, nascent insulin-containing secretory granules are targeted to lysosomes for degradation in a PKD-dependent manner [78].

On the other hand, PKD is also necessary to maintain the constitutive flow of secretory cargoes exiting the Golgi [30,31,58]. In polarized cells, PKD activity is necessary for the secretion of cargoes destined to the basolateral membrane (such as ß1-integrin or E-cadherin), but not to the apical membrane (such as gp135) [26]. In general, different direct TGN-to-PM transport carriers have been shown to require PKD activity for their formation. One of them is CARTS (see Section 3 below). Another one is transport carriers of the CtBP1/BARS (C-terminal-binding protein 1/brefeldin A ADP-ribosylation substrate)-dependent pathway [79,80], which mediates the export of cargoes such as VSV-G [37,81]. Although these transport carriers have a shared machinery (e.g., PKD), they can be distinguished based on some regulatory components and in the cargoes they contain (Table 1). It is also important to mention the role of the small GTPase Rab6 in export from the TGN. Rab6 regulates the export of diverse cargo proteins from the TGN to the cell surface, some of which are PKD-dependent cargoes, such as VSV-G (a cargo of CtBP1/BARS pathway) or ß1 integrin [49,50,53,82] (see also Table 1). Since CARTS contain Rab6 (see Section 3.1 below), this pathway is also supposed to be Rab6-dependent.

CtBP1/BARS is in a complex together with phosphatidylinositol 4-kinase IIIß (PI4KIIIß) and a 14-3-3γ dimer. The stability of this complex is dynamically maintained by two protein phosphorylation reactions. First, PKD phosphorylates PI4KIIIß ([83], see Section 2.5 below), which reinforces the interaction between PI4KIIIß and 14-3-3γ [84]. Second, PAK1 kinase phosphorylates CtBP1/BARS to also enhance the interaction between this latter protein and 14-3-3γ [81]. The complexed CtBP1/BARS activates the lysophosphatidic acid acyltransferase δ (LPAATδ), a lipid modifying enzyme that converts lysophosphatidic acid into phosphatidic acid [38,85,86], a reaction proposed to facilitate membrane scission [38,81,87,88].

Importantly, PKD-dependent membrane fission of CtBP1/BARS carriers is independent of dynamin-2 activity [37], and so possibly is also the fission of CARTS. This contrasts to other PKD-independent export routes from the TGN, such as ARF1-positive tubular carriers [89] or clathrin-coated vesicles that require dynamin-2 for fission [15]. PKD therefore likely mediates membrane scission by a mechanism that is different from classical dynamin-induced fission [87,90].

### 2.4. PKD Recruitment to the TGN and Activation

PKD is recruited by its N-terminal C1 domains to the TGN membranes, where it regulates transport carrier biogenesis for protein secretion. In particular, it is recruited by binding to DAG through its C1a domain and also to ARF1 by its C1b domain [25,32,62,73]. The C1a domain is necessary for TGN targeting of PKD and, when expressed in cells, the single C1a domain displays TGN localization, suggesting that this domain is also sufficient for PKD targeting to the TGN [32,62]. However, when the C1b domain is deleted from the wild type protein, or when a point mutant that abolishes binding to ARF1 is introduced, the C1a-domain-containing PKD mutant remains cytosolic [73], suggesting a more complex multi-factorial recruitment mechanism of the full-length protein. Interestingly, as we have already mentioned, ARF1 is not required for ilimaquinone-induced Golgi vesiculation [22], suggesting that the ARF1-PKD interaction likely plays a fine-tuning rather than a direct role in the regulation of the fission reaction.

Once at the TGN, PKD can be activated by protein kinase C (PKC) and trimeric G protein subunits Gßγ [22,23] (Figure 2a). Particularly, PKD is phosphorylated in its activation loop (located in the kinase domain) by DAG-activated PKC members, such as the Golgi-localized PKCη [31,58,91,92]. PKC-mediated phosphorylation activates PKD by increasing its inherently low basal kinase activity [58]. Besides PKC, other PKD activators, such as increased intracellular Ca^2+^ levels [93], have been reported. In addition to this, activated PKD can also be auto- and *trans*-phosphorylated, which is thought to serve as a kinetic means to control (reduce) the time that PKD is fully activated [58].

Interestingly, PKD dimerization is essential for its activation and therefore its catalytic activity [70,71]. PKD N-terminal ULD mediates its homo- and hetero-dimerization [27,70,71]. Aicart-Ramos et al. proposed a model in which PKD is auto-inhibited and monomeric in the cytosol, and only upon DAG-mediated recruitment to the TGN, PKD is able to form dimers. These dimers undergo *trans*-phosphorylation within the kinase domains, which ultimately leads to the dissociation of the dimeric kinase domains and thereby full activation of dimeric, membrane-associated PKD [71]. Remarkably, expression of a non-dimerizable PKD-KD mutant had only a very mild inhibitory effect in protein secretion [71], thereby providing an explanation for the dominant negative effect of PKD-KD expression [70,71].

### 2.5. PKD Phosphorylates Substrates at the TGN

Active PKD localized at the TGN membranes can phosphorylate a number of different substrates to mediate constitutive protein secretion (Figure 2a). Common to those that are involved in constitutive protein export from the TGN is that they are either lipid-metabolizing enzymes or have some inherent connection in Golgi lipid homeostasis (Figure 2b).

*PI4KIIIß:* PKD phosphorylates PI4KIIIß, triggering its activation [83]. Active PI4KIIIß phosphorylates the lipid phosphatidylinositol (PI) in the 4-position to yield PI 4-phosphate (PI4P). Interestingly, it has been suggested that ARF1 plays a determinant role in the localization and activation of PI4KIIIß, hence underscoring the interconnection amongst all these components associated to PKD-mediated transport carrier formation [17].*CERT:* PKD phosphorylates the ceramide transport protein (CERT) [94]. CERT localizes to ER-Golgi membrane contact sites (MCSs), and it was the first lipid transfer protein (LTP) found to shuttle lipids between two distinct membranes without a vesicular intermediate [95]. CERT contains a canonical PH domain, which has specificity for binding to PI4P and hence targets this protein to the *trans*-Golgi/TGN membranes; an FFAT (two phenylalanines in an acidic tract) motif for interaction with the ER-localized vesicle-associated membrane protein-associated proteins (VAPs); and a steroidogenic acute regulatory protein-related lipid transfer (START) domain, which is responsible for the directional transfer of ceramide from the ER to the Golgi membranes [95]. Upon phosphorylation by PKD, CERT does not bind PI4P and is therefore released from the TGN, which supposedly improves the efficiency of non-vesicular lipid transfer by enhancing its dynamics at ER-Golgi MCSs.*OSBP:* Another substrate of PKD at ER-Golgi MCSs is the oxysterol-binding protein (OSBP) [96]. OSBP is a member of the OSBP-related protein (ORP)/oxysterol-binding homology (Osh) family of sterol sensor proteins. Similar to CERT, OSBP contains a PH domain that binds PI4P and also to ARF1-GTP, thus targeting OSBP to the late Golgi membranes; as well as an FFAT motif for *trans* ER anchoring through VAPs. In addition, OSBP contains an OSBP-related domain (ORD), which acts as a bidirectional lipid transfer domain that catalyzes the transport of cholesterol from the ER to the *trans*-Golgi membranes, and the reciprocal transport of PI4P from the Golgi to the ER membrane [97]. Mechanistically, there is experimental evidence suggesting that OSBP can function as a ferry-bridge protein, meaning that it combines its tethering function—by linking the ER and Golgi membranes together—with its lipid transfer function—by shuttling lipids between these two membranes [98]. Importantly, this ferry-bridge mode of action has also been proposed for other proteins with a similar domain structure, such as CERT [98]. Likewise to CERT, PKD-mediated phosphorylation of OSBP releases it from PI4P at the TGN for efficient lipid transfer [17,96].

## 3. CARTS Are PKD-Dependent Transport Carriers from the TGN to the Cell Surface

### 3.1. Isolation and Identification of CARTS

CARTS were immuno-isolated from HeLa cells to characterize transport carriers that mediate the PKD-dependent constitutive secretion pathway from the TGN to the cell surface [39]. For this purpose, TGN46-containing carriers were generated in a PKD-dependent manner by an in vitro system where permeabilized HeLa cells were incubated with ATP-regenerating system and rat liver cytosol, and immuno-isolated with an anti-TGN46 antibody recognizing the cytosolic tail of TGN46. Mass spectrometry analysis of the isolated carriers revealed that CARTS contain a number of secretory and PM-specific cargoes, as well as synaptotagmin II and myosin II. Among these proteins, the secretory cargo pancreatic adenocarcinoma upregulated factor (PAUF) was found to be a suitable marker for CARTS because it is highly enriched in these transport carriers. Subsequent analyses using PAUF as a CARTS marker revealed that they also contain the small GTPases Rab6a and Rab8a, and that PKD activity is required for PAUF secretion and CARTS biogenesis. As described in Section 2.3, CARTS are supposed to be a subclass of PKD-dependent carriers and other classes of carriers exist for PKD-dependent constitutive and regulated secretion.

### 3.2. Features of CARTS Pathway

CARTS are different from carriers that transport collagen I and VSV-G from the TGN to the PM (actually, the size of CARTS range from 100 to 250 nm in diameter that is too small to transport collagens) [39]. So far, comparison between TGN export routes mediated by VSV-G-containing carriers and CARTS, respectively, has revealed molecular machineries specific to the latter, as follows.

Kinesin-5/Eg5 is specifically required for transport of CARTS on microtubules [40]. Kinesin-5/Eg5 is a plus end-directed bipolar kinesin, which is known to cross-link and slide anti-parallel microtubules to assemble a bipolar spindle in mitosis [99,100,101,102,103,104]. Therefore, it is plausible that in nonmitotic cells this protein promotes CARTS transport from the TGN toward the cell periphery by sliding anti-parallel microtubules. In pancreatic β cells, Golgi-derived microtubules form a dense non-radial meshwork which limits dwelling of insulin granules at the cell periphery and negatively regulates insulin secretion [105]. Kinesin-5/Eg5 might control formation of such microtubule-based structures for CARTS migration.

Myosin II was identified as a component of CARTS that is required for PAUF secretion, but not for CARTS biogenesis at the TGN [39]. Although myosin II has been shown to mediate membrane fission of Rab6-positive carriers from Golgi membranes and promote VSV-G transport to the cell surface [50,82], it is likely that myosin II is not required for CARTS biogenesis, but rather important for migration of CARTS through the cortical actin for fusion with the PM [39]. This example highlights how the same molecular machinery can control separate steps in the biogenesis of distinct transport carriers [106].

An electron microscopy-based analysis of CARTS morphology suggested that CARTS lack cytoplasmic coats, as common to most TGN-derived transport carriers [39,107]. In general, cytoplasmic coats provide a means to coordinate cargo sorting mediated by a signal sequence with carrier budding and membrane fission. Therefore, understanding how these events are coordinated during CARTS biogenesis emerged as a fundamental question. Increasing evidence to date strongly suggests that lipid metabolism at ER-Golgi MCSs plays a key role [41,108] (See also Section 4.1). Double knockdown of two isoforms of VAP (VAP-A and VAP-B) inhibited DAG-dependent recruitment of PKD to the TGN membranes and subsequent membrane fission for CARTS biogenesis [108]. Similarly, double knockdown of CERT and OSBP or overexpression of an OSBP mutant that immobilizes ER-Golgi MCSs inhibited CARTS biogenesis, suggesting that lipid transfer at ER-Golgi MCSs is required for CARTS biogenesis. Importantly, in these conditions, processing of the CARTS cargo PAUF was also inhibited. This could be attributed to glycosylation defects as a result of disruption of TGN functional enzymatic domains whose organization requires sphingomyelin (SM) homeostasis [109,110]. SM and cholesterol assemble into liquid-ordered membrane nanodomains that are segregated from other lipids and thus can function as a platform for specific proteins involved in glycosylation, cargo sorting, and transport carrier biogenesis [42,43,109,110,111,112,113,114,115,116]. Interestingly, CARTS are positive for the SM reporter protein EQ-SM and contain FM4-GPI, a GPI-AP known to associate with cholesterol- and sphingolipid-enriched detergent-insoluble membranes [41]. Altogether, CARTS biogenesis presumably depends on at least two signaling pathways: (i) DAG-dependent recruitment of PKD; and (ii) organization of SM- and cholesterol-enriched nanodomains at the TGN (Figure 3).

Double knockdown of VAP-A and VAP-B inhibits both VSV-G export from the TGN and CARTS biogenesis [108,117], suggesting that lipid transfer at ER-Golgi MCSs is commonly required for these two routes. However, our recent paper has demonstrated that sterol regulatory element-binding protein (SREBP) cleavage activating protein (SCAP), a cholesterol sensor in the ER membrane, specifically promotes CARTS biogenesis at ER-Golgi MCSs [41]. The detailed molecular mechanism is described in Section 5.2.

## 4. Lipid Requirements for PKD-Mediated CARTS Formation

### 4.1. ER-Golgi MCSs Are a Platform for Non-Vesicular, Bi-Directional Trafficking of Lipids

As described above, a number of ER-Golgi MCS proteins, including VAPs, CERT, OSBP, and SCAP, are necessary for CARTS biogenesis [41,108]. Interestingly, super-resolution nanoscopy showed that CARTS seem to form in TGN regions that are separate but adjacent to ER-Golgi MCSs [41]. How ER-Golgi MCSs mechanistically mediate CARTS biogenesis still remains to be fully elucidated [118], but there is experimental evidence that suggests a prominent role for lipids in this process.

ER-Golgi MCSs have been identified as platforms for the fast and directional transport of lipids between these two organelles, and hence emerge as key organizers of lipid homeostasis in the secretory pathway [118,119,120] (Figure 3). These contact sites are established by the ER-localized VAP proteins. VAPs are integral membrane proteins with a cytosolic major sperm protein (MSP) domain, which binds in *trans* to proteins containing an FFAT motif, thereby tethering the ER with other organelles [121]. In particular, at the ER-Golgi interface, VAPs interact with the PKD substrates CERT and OSBP (see Section 2.5), as well as with Nir2 and four-phosphate adaptor protein 1 and 2 (FAPP1 and FAPP2) [119].

Nir2 is a peripheral protein that contains a PI transfer protein (PITP) domain that has been proposed to allow Nir2 to shuttle PI from the ER to the Golgi membranes. Nir2 also contains an FFAT motif for ER-Golgi MCS localization through the interaction with the ER-localized VAPs [117,122].

FAPP1 and FAPP2 show 90% similarity at the N terminus, but differ at the C terminus because only FAPP2 contains a glycolipid transfer protein homology (GLTPH) domain [123]. Both FAPP1 and FAPP2 contain a PH domain for binding to PI4P and ARF1-GTP at the *trans*-Golgi membranes, and an FFAT motif that potentially leads to interaction with VAPs at the ER membrane [124]. FAPP2, but not FAPP1, is an LTP that specifically binds and transfers glucosylceramide not from the ER, but from the *cis*-Golgi membranes—where this lipid is synthesized—directly towards the *trans*-Golgi membranes [119,125,126]. Recent work reported that FAPP1 functions as a PI4P detector/adaptor to control Golgi PI4P levels ([127], See Section 5.2).

Taking all this information together, it is clear that ER-Golgi MCSs are fundamental to maintain the lipid homeostasis at the Golgi membranes. Whereas transport carrier-mediated lipid trafficking from the ER to the *trans*-Golgi membranes is a relatively slow process (ER-to-TGN trafficking of the fastest cargoes takes longer than 15 min [128,129]), MCSs can serve as platforms for a much faster and specific supply of lipids to the *trans*-Golgi membranes [130]. Moreover, LTP-mediated lipid delivery is spatially restricted at the MCS region, thus creating a transient and local enrichment of certain lipid species at specific membrane areas. Whether and how this fast and localized lipid transfer has a functional relevance for transport carrier formation at the TGN still remains to be fully understood [41].

### 4.2. The Cycle of PI4P at the ER-Golgi Interface

As described above, ER-Golgi MCSs are sites enriched in proteins of a high non-vesicular lipid transfer activity. Interestingly, proteins such as CERT and OSBP localize to the late Golgi membranes (*trans*-Golgi/TGN cisternae) by binding to the Golgi-enriched phosphoinositide PI4P. How are Golgi levels of PI4P controlled? On the one hand, PI4 kinases are proteins with a catalytic activity to specifically phosphorylate PI in the 4-position to generate PI4P. Mammalian cells express four of these kinases: two of which are type II (PI4KIIα and PI4KIIß) and two type III (PI4KIIIα and PI4KIIIß) [131,132]. Two of those lipid kinases (PI4KIIIß and PI4KIIα) clearly localize to the Golgi membranes [133], with PI4KIIIβ only localized at the Golgi and PI4KIIα localized at the Golgi and endosomes [134].

On the other hand, PI4P is transported without a vesicular intermediate by OSBP from the *trans*-Golgi membranes to the ER. At the ER, the PI4P-specific phosphatase Sac1 rapidly dephosphorylates PI4P to generate PI thereby creating an ER-Golgi PI/PI4P cycle [97,98].

In summary, our current understanding is that ER-Golgi MCSs control the metabolic flow and homeostasis of the phosphoinositide PI4P, sphingolipids, and cholesterol (Figure 2b). As such, it is reasonable to propose that ER-Golgi MCSs maintain the identity of the Golgi complex by keeping steady PI4P levels (through OSBP and Sac1) and establishing a sphingolipid (through CERT) and cholesterol (through OSBP) gradient along the membranes of the secretory pathway.

### 4.3. Sources of DAG at the TGN for PKD Recruitment and Activation

As we have already mentioned, DAG is crucial to recruit and active PKD at the Golgi, and consequently for PKD function in CARTS biogenesis. DAG is a neutral, uncharged glycerolipid that consists of two fatty-acid (acyl) chains ester-linked to a glycerol molecule. DAG metabolism is complex, and cells use a variety of routes to maintain the homeostasis of this lipid molecule [135] (Figure 2b).

De novo DAG biosynthesis starts with the acylation of glycerol-3-phosphate or dihydroxyacetone-3-phosphate into lysophosphatidic acid (LPA), and proceeds with the subsequent acylation of LPA into phosphatidic acid (PA). PA can then be dephosphorylated by a PA phosphohydrolase (PAP) to produce DAG. In parallel to this de novo pathway, DAG can also be formed as a metabolic product or by-product of different lipid reactions. Hence, DAG can be generated from glycerophospholipids (such as phosphatidylcholine (PC), PI, phosphatidylserine (PS), or phosphatidylethanolamine (PE)) directly by the action of phospholipase C (PLC). In particular, Golgi-localized PLCß3 and PLCγ1 cleave the headgroup of phosphoinositides (most likely PI4P) to generate DAG for export from the TGN [136,137] (see below).

Alternatively, DAG can also be produced by means of a double reaction, which uses PA as an intermediate. This double reaction consists of the sequential action of phospholipase D (PLD)—which converts a glycerophospholipid (such as PC or PI) into PA—after which PAP produces DAG out of PA. In addition, DAG is also formed as a by-product of the sphingomyelin synthase (SMS) enzyme-mediated reaction (see below, and Figure 2b).

The levels of DAG in Golgi membranes isolated from HeLa cells have been measured by lipid mass-spectrometry [109], representing ~1–2 mol% of total lipid levels. Interestingly, boosting the SMS-mediated generation of DAG by exogenous addition of short-chain ceramide leads to a ~30% relative increase in the Golgi levels of DAG [109], which parallels a sharp and fast (~15 min) activation of PKD [116].

Having no polar headgroup makes DAG a highly non-polar lipid. This, together with the fact that it is uncharged, makes DAG a lipid with a very high flip-flop rate [138]. In addition, DAG has a conical effective molecular shape. Although cone-shaped lipids are able to induce monolayer bending [139], they only induce bilayer bending when distributed asymmetrically between the two monolayer leaflets. Therefore, DAG is not likely to be a very powerful membrane curvature generator in a cellular context [115].

Is there any specific metabolic route for DAG production that is more relevant for PKD-mediated transport carrier formation? On the one hand, the PI-PLC family member PLCß3 is required for Gßγ-induced Golgi fragmentation and VSV-G export from the TGN, possibly in a CtBP1/BARS-dependent manner [136]. Gßγ translocation to the TGN membrane enhances the activity of TGN-localized PLCß3, which in turn cleaves the headgroup of phosphoinositides (most likely PI4P) to generate DAG for PKD recruitment and activation. In addition, another PLC isoform, PLCγ1, has been shown to mediate DAG production for PKD activation and VSV-G export from the TGN [137] (Figure 2b).

On the other hand, PKD modulates the activity of CERT via phosphorylation (Section 2.5), and CERT shuttles ceramide from the ER to the TGN. CERT-transported ceramide, together with PC, is converted into SM and DAG by the Golgi-localized SMS enzymes (SMS1 and SMS2), and this overall metabolic route is fundamental for transport carrier formation at the TGN [109,115,116]. Importantly, this metabolic reaction is coupled to secretory pathway Ca^2+^ ATPase 1 (SPCA1)-mediated Ca^2+^ entry into the TGN [43], which ultimately controls the sorting of a subset of cargo proteins [17,19] (see Section 4.4).

Finally, ARF1, which is a binding partner of PKD [73], can activate PLD [140], the enzyme responsible for PC conversion into PA [141]. PLD activity also seems to be required for ilimaquinone-induced Golgi vesiculation [142] and for ER-to-Golgi trafficking [143]. How PLD mediates specific PKD-dependent export routes from the TGN still remains to be completely elucidated.

In summary, given that VSV-G is a CtBP1/BARS-specific cargo that is excluded from CARTS, we hypothesize that PLC-mediated DAG formation is important for this particular export route. By contrast, we propose that CARTS biogenesis requires SMS-mediated DAG formation for local PKD recruitment and activation at ER-Golgi MCSs [41,108]. Although we still do not know how PKD can differentially regulate and distinguish between different types of carriers, an appealing hypothesis is that different DAG sources define different PKD-dependent export routes from the TGN (Figure 2). It will be important in the future to simultaneously image by super-resolution nanoscopy the budding sites of the different PKD-dependent transport carriers to test whether different TGN subdomains—possibly enriched in different DAG subspecies and/or PKD effectors—serve as specific platforms for the biogenesis of different PKD-dependent transport carriers. In addition, lipidomics of isolated transport carriers will provide important clues on the lipid metabolic pathways controlling each export route.

### 4.4. SM Metabolism and Cab45-Mediated Secretory Cargo Sorting at the TGN

In 2009, von Blume et al. found that the actin-severing proteins actin depolymerizing factor (ADF) and cofilin-1 are required for the sorting of a subset of cargo proteins at the TGN [46]. Subsequent work identified the TGN-localized calcium pump SPCA1 as a downstream component of ADF/cofilin-mediated cargo sorting [44]. SPCA1 Ca^2+^ pumping activity, which is stimulated by recruitment of F-actin via cofilin1 to this protein, is necessary to create regions of high local Ca^2+^ concentration in the TGN lumen, which favor the oligomerization of Cab45, a soluble calcium-binding protein localized in the TGN lumen at steady state [45,47]. In addition, phosphorylation of Cab45 by the Golgi-localized soluble protein kinase Fam20C is important for Cab45-mediated cargo sorting and protein secretion [144]. Notably, Cab45 was shown to be abnormally secreted in cells depleted of either ADF/cofilin or SPCA1, thereby underscoring the importance Cab45 as a Golgi-localized sorting factor [44,46]. Another cargo molecule that is not properly secreted upon abrogation of the Ca^2+^ homeostasis in the TGN is lysozyme C [44,45,46], which happens to also be a cargo of the CARTS pathway [39] (Table 1). Hence, it is very likely that Ca^2+^-mediated cargo sorting is an important mechanism for cargo selectivity into CARTS [15,19] (Figure 2a).

Importantly, in response to RhoA activation, PKD negatively regulates the activity of cofilin through phosphorylation (inactivation) of slingshot, a protein phosphatase that dephosphorylates (activates) cofilin [145]. Although this mechanism was shown to support actin remodeling at the leading edge of migrating tumor cells, it might also be playing a similar role at the TGN for cargo sorting coordinated with membrane fission for transport carrier biogenesis. Notably, a similar cofilin-mediated cargo sorting pathway has also been described in *S. cerevisiae* [146]. Because no PKD ortholog has been found in yeast, calcium-mediated cargo sorting at the TGN has possibly a broader range of action than that of CARTS/PKD-mediated protein secretion.

Finally, work from the von Blume and Burd labs, respectively, indicated that Golgi SM enhances SPCA1-mediated Ca^2+^ pumping, and that Ca^2+^-induced Cab45 oligomerization helps package cargo proteins into SM-rich transport carriers [42,43,48] (Table 1). It is therefore clear that there is a key connection between SM metabolism and this specific pathway of cargo sorting and export from the TGN, similarly to CARTS pathway (Figure 2a).

## 5. Signaling at the Golgi Membranes for the Regulation of Protein Secretion

Thus far we have highlighted a number of molecular players that contribute to TGN export in general, and to CARTS formation, particularly in the context of PKD substrates. We have also reviewed a number of upstream activators of PKD (Section 2). However, a key question in the field that still remains open is how membrane trafficking is regulated in response to physiological needs [3,4,5,118,147]. We discuss here different signaling pathways that transduce cellular cues into signals to control the secretory flow. The initiating signals can be environmental signals external to the cell, such as metabolic needs, stress signals, nutrient availability, or mechanical cues [4,148] (Figure 2a). Alternatively, signals arising from within the cell, such as variations in membrane traffic load or cholesterol availability, can also act as igniting factors to regulate protein secretion [5] (Figure 2a).

### 5.1. GPCR-Mediated Signaling

GPCRs (G protein-coupled receptors) are seven-pass-transmembrane cell surface receptors that, upon ligand binding, activate G proteins to mediate a number of different intracellular signaling pathways [149]. The G proteins that mediate GPCR signaling form heterotrimeric complexes composed of α, β, and γ subunits, and the latter two are often found forming a βγ subcomplex. In particular, GPCR signaling is involved in the maintenance of Golgi structure and function [5,150], mediated by different Gα subunits that have been localized to the Golgi membranes [5,151].

On the one hand, as aforementioned, PKD activation occurs by the release of the βγ complex upon G protein activation [22], which leads to PLCß3 activation and, downstream of this, PKC and PKD activation at the TGN for transport carrier formation [3,31]. In the context of regulated secretion, it was shown that translocation of G protein βγ complex to the Golgi mediates insulin secretion in a PKD- and PLCß-dependent manner [152].

On the other hand, a conceptually similar G protein-dependent signaling pathway, mediated by the KDEL receptor (KDELR), takes place on the Golgi. KDELR is an ER-Golgi localized transport receptor with a similar topology as that of GPCRs. The canonical function of KDELR is to bind KDEL sequence-containing proteins (mostly chaperones) that escaped the ER and reached the Golgi complex, and escort them back to the ER. However, in recent years, alternative functions have been ascribed to this protein [3,5,153]. Amongst these novel functions, it has been suggested that the KDELR can act as a signaling hub for the regulation of membrane trafficking to maintain Golgi homeostasis [5].

In order to maintain Golgi membrane homeostasis during large waves of cargo being transported from the ER to the Golgi, the Golgi needs to first sense and then balance the excess received membrane by increasing the rate of transport carrier formation. Within this conceptual framework, KDELR is proposed to act as a key receptor that regulates Golgi trafficking [3,4,5]. First, enhanced cargo flow from the ER to the Golgi complex leads to the Gα_q_-dependent activation of the Src tyrosine kinase at the Golgi membranes [154,155], which ultimately leads to dynamin-2-dependent anterograde transport from the TGN [156]. Second, cargo-bound KDELR also stimulates Gα_s_ to activate protein kinase A (PKA), which in turn upregulates the transcription of proteins involved in retrograde Golgi-to-ER trafficking [157]. Hence, Luini and colleagues proposed the existence of a Golgi control system, with the KDELR as one of its central pieces. This control system would help maintain Golgi homeostasis in response to membrane flow [5].

In a recent preprint, Di Martino et al. showed that the orphan GPCR GPRC5A is a TGN-localized cargo sensor [158]. Upon arrival of basolateral-directed cargoes (such as albumin, VSV-G, or TNFα), GPRC5A in complex with Gαi3 and Gβγ triggers the activation of PLCß3, PKC and PKD for basolateral cargo export. As reviewed above, the involvement of PLCß3 suggests that this PKD-mediated export pathway is that of CtBP1/BARS carriers and not CARTS. How some cargoes specifically bind and/or activate GPRC5A remains an open question. In summary, this work presents a novel autoregulatory mechanism, which the authors named as the ARTG pathway, for autoregulation of TGN export [158]. A similar autoregulatory pathway at the level of ER export has also been unraveled [159].

### 5.2. Sac1-Mediated Signaling

The activity of Sac1, a transmembrane PI4P phosphatase, is controlled by multiple pathways to regulate Golgi export. Sac1 has been demonstrated to link growth factor signaling to lipid signaling at the Golgi [160]. In serum-starved quiescent cells, Sac1 forms oligomers interacting with the COPII component Sec23, leading to ER exit and Golgi accumulation of Sac1. As a result, Sac1 depletes Golgi PI4P and down-regulates anterograde transport from the Golgi. On the other hand, activated p38 MAPK pathway under mitogen stimulation induces dissociation of Sac1 oligomers and causes COPI-dependent retrieval of Sac1 to the ER, thus releasing the brake on Golgi export.

In neurons, Sac1 activity is down-regulated by carnitine palmitoyltransferase 1C (CPT1C), a sensor of malonyl-CoA, for Golgi export of the major AMPA receptor subunit GluA1 [161]. Malonyl-CoA is synthesized from acetyl-CoA by acetyl-CoA carboxylase (ACC), as an intermediate in the de novo synthesis of long-chain fatty acids. Since malonyl-CoA levels are modified by different metabolic stress conditions, including glucose depletion, the CPT1C-Sac1 axis is suggested to underlie regulation of cell surface expression of GluA1 reflecting nutrient conditions. Given that the activity of ACC is negatively controlled by the master energy sensor AMP-activated protein kinase (AMPK), the CPT1C-Sac1 axis seems to be downstream of the AMPK pathway. Although Sac1 has been proposed to translocate to the Golgi and act in *cis* on Golgi PI4P pool in quiescent cells [160], inhibition on malonyl-CoA synthesis promoted formation of ER-Golgi MCSs and recruitment of CPT1C (no malonyl-CoA binding)-Sac1 complex to the ER side of the MCSs. At the same time, CPT1C inhibition of Sac1 is released causing GluA1 retention at the TGN. These results favor the idea that Sac1 acts in *trans* [161]. So far, how Sac1 functions in controlling Golgi PI4P levels remains controversial. Probably, multiple modes of Sac1 activity (including the description below) coexist as distinct modes, but in an integrated fashion.

FAPP1 is ubiquitously expressed and acts as a PI4P detector and as an activator of Sac1 to regulate Golgi PI4P levels [127]. FAPP1 localizes to ER-Golgi MCSs where it forms a complex with Sac1 and VAPs to promote in *trans* activity of Sac1. Considering that the intramembrane distance at ER-Golgi MCSs shows a heterogeneous distribution (in the range from 5 to 20 nm [162]), in *trans* activity of Sac1 is supposed to occur only at the tighter MCSs including the Sac1-FAPP1-VAPs complex [127]. Knockdown of FAPP1 increases Golgi PI4P and specifically promotes the export of ApoB100 from the TGN, suggesting the role of this protein as a negative regulator of selected carrier biogenesis at the TGN [127].

As described in Section 4.2, Sac1 is also supposed to function in *cis* at the ER membrane to provide a driving force for cholesterol transport at ER-Golgi MCSs by hydrolyzing PI4P which is reciprocally transported from the *trans*-Golgi/TGN to the ER [97,98]. Exploring Sac1-interacting proteins at ER-Golgi MCSs revealed that the SCAP-Sac1 axis promotes CARTS biogenesis in an ER cholesterol-dependent manner [41]. SCAP is a polytopic membrane protein well known to play a key role in cholesterol metabolism [163]. In response to cholesterol deprivation, SCAP escorts the membrane-bound SREBPs into COPII vesicles for their ER export. At the Golgi membranes, SREBPs are cleaved and their transcriptionally active domain enters the nucleus to promote gene expression for cholesterol synthesis and uptake. Conversely, upon high cholesterol levels in the ER membrane, cholesterol-bound SCAP sequesters SREBPs at the ER through interaction with the integral ER membrane protein Insig. Under cholesterol-fed conditions, a part of the ER-localized pool of SCAP/SREBP complex interacts via Sac1 with a VAP-OSBP complex at ER-Golgi MCSs to promote PI4P turnover and cholesterol transport, leading to CARTS biogenesis at the TGN (Figure 3) [41]. In this model, Sac1 positively controls CARTS biogenesis, by contrast to the other models described above. Given that the cell surface expression of GluA2 and GluN2A is insensitive to inhibition on malonyl-CoA synthesis [161] and that secretion of some cargoes such as albumin and α1-anti-trypsin [127] is insensitive to FAPP1 knockdown, different classes of transport carriers will be sensitive to different PI4P levels or pools in the TGN membranes.

### 5.3. Mechanical Signals

Mechanical properties of the ECM are crucial not only for tissue structure but also regulation of cellular adhesion and migration, proliferation, and differentiation. Cells sense the mechanical properties through integrin receptors and adjust actomyosin contractility for measurement. Interestingly, redistribution of SCAP/SREBP complex to the Golgi and subsequent activation of SREBPs are controlled by ECM mechanical cues in a cholesterol-independent manner [164]. A soft ECM microenvironment (a condition of low actomyosin contractility) inhibits activity of Lipin-1 phosphatidate phosphatase and decreases Golgi DAG levels, followed by ARF1 inactivation and SCAP/SREBP accumulation to the Golgi, thus inducing synthesis of cholesterol and neutral lipids. Considering the role of the ER pool of SCAP/SREBP in CARTS biogenesis [41], this condition could possibly inhibit CARTS biogenesis. By contrast, ECM stiffness destabilizes microtubules and releases/activates GEF-H1, the Rho guanine nucleotide exchange factor (GEF) whose activity is further regulated by GPCRs [165,166]. GEF-H1 has been shown to activate PKD through activation of RhoA and its effector PLCε that produces DAG from PI4P at the TGN [52]. Importantly, this pathway regulates formation of Rab6/Rab8-positive carriers, which share some similarities to CARTS [15,39], for localized exocytosis of integrins and other cargoes at the immediate vicinity of focal adhesions [52,53,56].

Interestingly, GBF1, a GEF for ARF1 and ARF4, was also recently shown to mediate another mechanism for the regulation of the secretory pathway as a response to environmental needs [167]. In that article, the authors showed that GBF1 is activated downstream of AMPK in response to metabolic changes (extracellular glucose levels), and that this can control the secretion of von Willebrand factor (VWF) and ECM proteins. Specifically, in the absence of GBF1, VWF export from the TGN slows down. Notably, AMPK, a kinase involved in glucose and lipid metabolism, phosphorylates GBF1 to control its secretory function at the Golgi. Although involvement of PKD or ER-Golgi MCSs in this pathway is not known, this example underlines another mechanism by which extracellular signals can control the flow of secretory cargoes exiting the Golgi.

## 6. Summary and Concluding Remarks

We have reviewed here our current understanding of how TGN-to-cell surface membrane trafficking and constitutive protein secretion is controlled, with a special emphasis on the mechanisms that regulate CARTS formation through PKD. It is worth pointing out that the term “constitutive secretion” might need to be revisited, since this particular form of secretion is also highly regulated, as we hope became clear from this review. The establishment of PKD as a master regulator of transport carrier formation at the TGN was followed by a number of breakthroughs that extended the role of PKD in regulating Golgi lipid homeostasis. Because no cytosolic coats have yet been found to assist in CARTS formation, the discovery of protein machinery involved in cargo sorting, membrane bending and carrier scission, downstream of PKD, remains a challenge. Indeed, lack of cytosolic coats also presents a challenge in defining the different trafficking routes from the TGN to the cell surface. As shown in Table 1, some cargoes are commonly included in the transport carriers listed, indicating overlapping of some machineries on the same transport carriers. We expect that future work combining cargo synchronization methods together with state-of-the-art fluorescence microscopy techniques with ultra-high spatial and temporal resolution will help reveal the in situ regulation of different TGN export routes. In addition, in vitro reconstitution of PKD-dependent transport carrier biogenesis can serve as a workbench for testing the precise role of the different molecular players involved. It is evident that highly complex, interdependent cellular signals, such as nutrient availability, stress conditions, and mechanical cues, are sensed by the trafficking machinery to adapt the secretory pathway in response to those physiological needs. Despite these challenges, the list of mechanisms and signaling molecules that sense and relay those signals to the secretory machinery, therefore upstream of PKD, is steadily growing. The study of the signal-regulated constitutive secretion by using physiologically-relevant cellular models as well as in vivo model systems will provide important clues to understand the roles of protein secretion in health and disease. We expect that future efforts along these lines of research will provide us with a more holistic view of the secretory pathway and protein secretion in the general context of cell and tissue biology.

## Figures and Tables

**Figure 1 cells-10-01618-f001:**
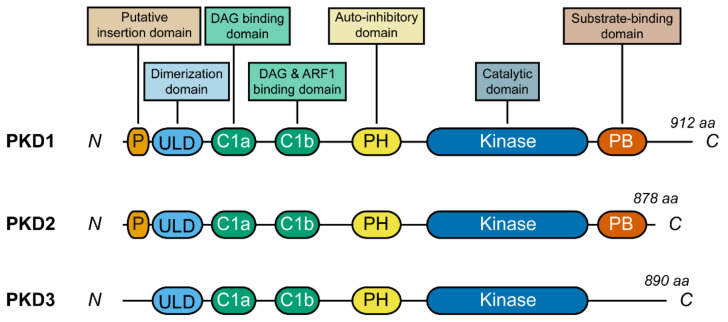
Schematic representation of human PKD domain structure. The three members of the human PKD family of protein kinases (PKD1, PKD2, and PKD3) have a similar domain structure. The major domains and their functions are depicted here. P: putative insertion domain; ULD: ubiquitin-like domain; C1a and C1b: cysteine-rich zinc-finger regions a and b; PH: pleckstrin homology domain; PB: postsynaptic density-95/discs large/zonula occludens-1 (PDZ)-binding domain.

**Figure 2 cells-10-01618-f002:**
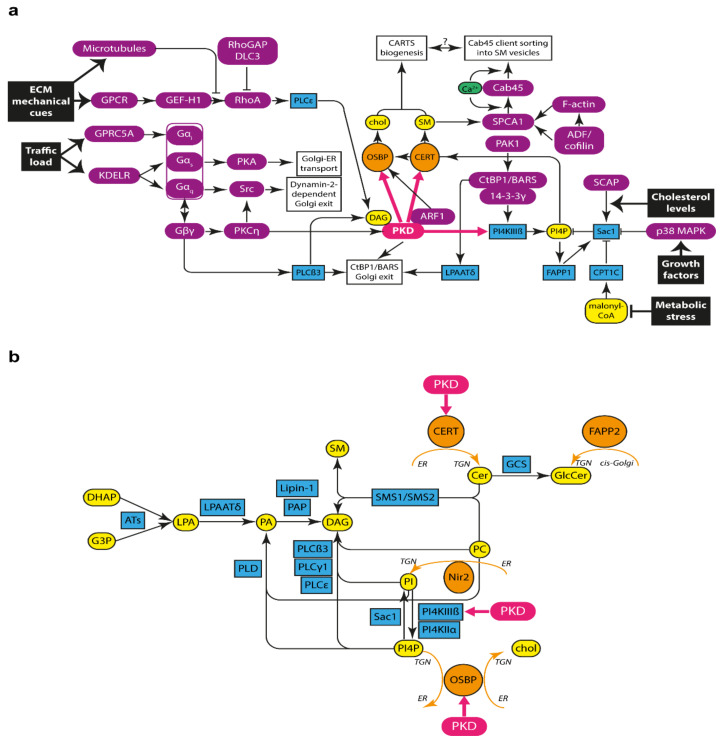
Lipid and protein signaling pathways that regulate Golgi lipid homeostasis and TGN-to-PM transport. (**a**) Signaling pathways leading to the activation of PKD (in pink) for the regulation of transport carrier biogenesis at the TGN. LTPs shown in orange, lipids in yellow, lipid metabolizing enzymes in blue, and other protein in dark violet. Upstream signals leading to regulation of the different pathways are shown as black text boxes, and the different Golgi export routes are depicted as white text boxes. (**b**) Lipid metabolic routes involved in DAG synthesis and Golgi lipid homeostasis. PKD shown in pink, LTPs shown in orange, lipids in yellow, lipid metabolizing enzymes in blue. Orange arrows indicate non-vesicular lipid transport, and pink arrows represent PKD-mediated phosphorylation. ATs: acyltransferases; cer: ceramide; CERT: ceramide transport protein; chol: cholesterol; DAG: diacylglycerol; DHAP: dihydroxyacetone-3-phosphate; FAPP2: four-phosphate adaptor protein 2; GlcCer: Glucosylceramide; GCS: GlcCer synthase; G3P: glycerol-3-phosphate; LPA: lysophosphatidic acid; LPAAT: LPA acyltransferase; OSBP: oxysterol-binding protein; PA: phosphatidic acid; PAK1: p21-activated kinase 1; PAP: PA phosphohydrolase; PC: phosphatidylcholine; PI: phosphatidylinositol; PI4K: PI 4-kinase; PI4P: PI 4-phosphate; PKA: protein kinase A; PKC: protein kinase C; PKD: protein kinase D; PLC: phospholipase C; PLD: phospholipase D; SM: sphingomyelin; SMS: SM synthase. See text for details.

**Figure 3 cells-10-01618-f003:**
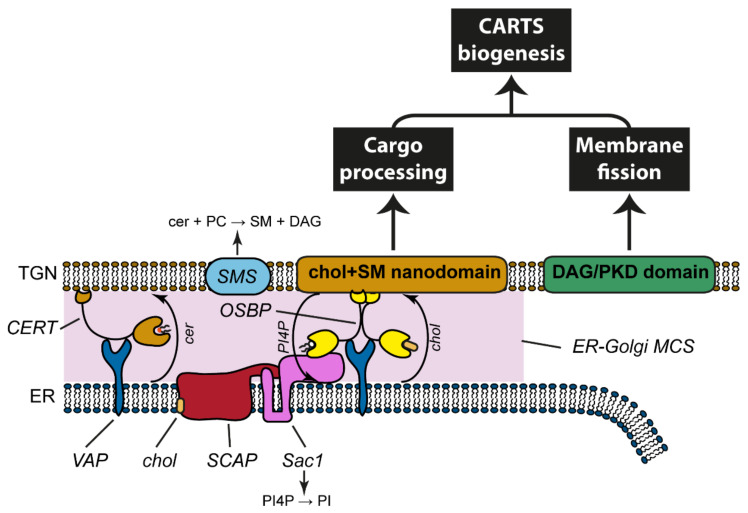
Lipid homeostasis at ER-Golgi MCSs for CARTS biogenesis. ER-Golgi MCSs serve as platforms for the non-vesicular transport of lipids and mediate the PI4P cycle at the ER-Golgi interface, as well as DAG, SM, and cholesterol homeostasis. cer: ceramide; chol: cholesterol; DAG: diacylglycerol; MCS: membrane contact site; PC: phosphatidylcholine; PI: phosphatidylinositol; PI4P: PI 4-phosphate; PKD: protein kinase D; SM: sphingomyelin; SMS: SM synthase. See text for details.

**Table 1 cells-10-01618-t001:** Protein cargoes in TGN-to-PM transport carriers. The transport carriers considered here do not necessarily describe completely different export routes, because they share some machinery and cargoes (see text for details). However, since these transport carriers have been defined and characterized as such in the literature, we describe them separately here. TM: transmembrane protein; MA: membrane-associated protein; GPI-AP: glycosyl-phosphatidylinositol-anchored protein.

Transport Carriers	Protein Name	Topology	Secretion Type	References
PKD-dependent carriers (General)	CD4	TM	Constitutive	[25]
	TGN46	TM	Constitutive	[25]
	Furin	TM	Constitutive	[32]
	VSV-G (*Basolateral PM*)	TM	Constitutive	[25,26]
	ss-HRP	Soluble	Constitutive	[27]
	Insulin	Soluble	Regulated	[33,34]
	Chromogranin A	Soluble	Regulated	[35,36]
	ß1-integrin (*Basolateral PM*)	TM	Constitutive	[26]
	E-cadherin (*Basolateral PM*)	TM	Constitutive	[26]
CtBP1/BARS-dependent carriers	VSV-G	TM	Constitutive	[37]
	Low-density lipoprotein (LDL) receptor	TM	Constitutive	[38]
	Human growth hormone (hGH)	Soluble	Constitutive	[38]
CARTS	TGN46	TM	Constitutive	[39]
	ss-HRP	Soluble	Constitutive	[39,40]
	PAUF	Soluble	Constitutive	[39]
	Lysozyme C	Soluble	Constitutive	[39]
	Synaptotagmin II	TM	Constitutive	[39]
	FM4-GPI (CD58)	GPI-AP	Constitutive	[41]
	EQ-SM (SM reporter protein)	MA	Constitutive	[41]
SM-rich transport carriers	EQ-SM (SM reporter protein)	MA	Constitutive	[42]
	FM4-GPI (CD58)	GPI-AP	Constitutive	[42]
	Cab45	Soluble	Constitutive	[43,44,45]
	ss-HRP	Soluble	Constitutive	[44,46]
	Lysozyme C	Soluble	Constitutive	[45,47]
	Cartilage oligomeric matrix protein (COMP)	Soluble	Constitutive	[45,47]
	Lipoprotein lipase	MA	Constitutive	[48]
	Syndecan-1	TM	Constitutive	[48]
Rab6-positive carriers (General)	VSV-G	TM	Constitutive	[49,50]
	Neuropeptide Y (NPY)	Soluble	Constitutive	[49,51]
	TNFα	Soluble	Constitutive	[52,53,54]
	Herpes Simplex virus 1 (HSV1) glycoproteins gD/gE	TM	Constitutive	[55]
	Collagen type X	Soluble	Constitutive	[53]
	CD59	GPI-AP	Constitutive	[53]
	Placenta alkaline phosphatase (PLAP)	GPI-AP	Constitutive	[53]
	ß1-integrin	TM	Constitutive	[56]
	Membrane type 1-matrix metalloproteinase (MT1-MMP)	TM	Constitutive	[57]

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
