# Peer review of "The PKD-Dependent Biogenesis of TGN-to-Plasma Membrane Transport Carriers"

_cells, 2021, doi:10.3390/cells10071618_

Round 1
Reviewer 1 Report
This review is an exciting and extensive report on the mechanism and regulation of a specific set of membrane traffic routes. Although the review is mainly focused on a limited number of specialized traffic pathways originated at the Trans Golgi Network, the conclusions and the models that can be inferred by discussing the feature of these extensively investigated membrane traffic steps can be of interest to a broader audience.
The references are updated and represent the current knowledge fairly. The review is, in general, well written, and there are not many reviews about this specific topic. Thus, even if the theme can appear to be of interest to a specialized audience, it is novel, and therefore I think this can legitimate the publication in "cells".
Major suggestions
The title is catching, but it gives the impression of a review reporting on all the TGN-based traffic steps. Please, consider to rephrase it to describe the content better.
Chapter 1.2 Multiple export pathways from the TGN. The authors state that it is not the review's aim to elaborate on all of the molecular determinants and pathways of cargo export from the TGN. However, I suggest revaluating this view and elaborate a little more on the chapter by briefly mentioning the essential protein machineries and pathways that compose other exit routes.
Connected to the above comments is Table 1. I think this is an interesting table listing essential components involved in the transport of a number of cargoes. By reading the table, I understand that the authors describe the main features of five distinct export routes. I am not fully up-to-date with this specific literature, but this is surprising to me. To my knowledge, many of the components described in the table are part of the same traffic routes, and I recall that there are no more than two/three exit routes directed to the PM. I have not been able to address my doubts by reading the rest of the manuscript. Thus, I suggest the authors to address this unclear issue.
I wonder whether the term “constitutive flow” can still be valid, as in reality the so-called constitutive secretion is finely regulated, as well detailed in this review. I am not suggesting a revision of the term, but just a second thought.
Minor points
Lanes 55-76; the statements are not supported by references.
Chapter 2.2 PKD Structure; lanes 144-154. The domain organization and the regulation of PKD are quite complex and not fully understood. I do not have a broad knowledge of the recent pertinent literature, but to my knowledge, even if the C1a domain is the major determinant for DAG binding and membrane recruitment, also the C1b domain can bind DAG, even if it appears to have a higher affinity for PMA. Finally, PKD1 and 2 contain a C-terminal PDZ domain responsible for interaction with several protein substrates, but this is not mentioned in the text. I also suggest evaluating the possibility of adding a schematic representation of the PKDs, as this protein family is a major regulator of carrier formation and is an important “transducer” of the signalling pathways described in this review.
Lane 168, the correct definition of BARS is: Brefeldin-A ADP-ribosylation substrate
Author Response
REVIEWER 1
This review is an exciting and extensive report on the mechanism and regulation of a specific set of membrane traffic routes. Although the review is mainly focused on a limited number of specialized traffic pathways originated at the Trans Golgi Network, the conclusions and the models that can be inferred by discussing the feature of these extensively investigated membrane traffic steps can be of interest to a broader audience.
The references are updated and represent the current knowledge fairly. The review is, in general, well written, and there are not many reviews about this specific topic. Thus, even if the theme can appear to be of interest to a specialized audience, it is novel, and therefore I think this can legitimate the publication in "cells".
We thank the reviewer for the positive evaluation of our review.
Major suggestions
The title is catching, but it gives the impression of a review reporting on all the TGN-based traffic steps. Please, consider to rephrase it to describe the content better.
We have changed the title to better describe the content of our review. The new title is: "The PKD-dependent biogenesis of TGN-to-plasma membrane transport carriers".
Chapter 1.2 Multiple export pathways from the TGN. The authors state that it is not the review's aim to elaborate on all of the molecular determinants and pathways of cargo export from the TGN. However, I suggest revaluating this view and elaborate a little more on the chapter by briefly mentioning the essential protein machineries and pathways that compose other exit routes.
We thank the reviewer for the suggestion. Although we still do not provide a full description of the TGN export mechanisms, we expanded Chapter 1.2 to include more information about the machinery involved in TGN export.
Connected to the above comments is Table 1. I think this is an interesting table listing essential components involved in the transport of a number of cargoes. By reading the table, I understand that the authors describe the main features of five distinct export routes. I am not fully up-to-date with this specific literature, but this is surprising to me. To my knowledge, many of the components described in the table are part of the same traffic routes, and I recall that there are no more than two/three exit routes directed to the PM. I have not been able to address my doubts by reading the rest of the manuscript. Thus, I suggest the authors to address this unclear issue.
We thank the reviewer for pointing this out, which was clearly not explained with enough detail in our previous submission. We expanded the text to hopefully provide with a more clear vision. The 5 "routes" described in Table 1 are not independent of each other, but most likely some of them can represent the same route. We now explain this in more detail also in the Table legend: "The transport carriers considered here do not necessarily describe completely different export routes, because the machinery is shared amongst some of them (see text for details). However, since they have been described and characterized as such in the literature, we refer to them as separate carriers."
I wonder whether the term “constitutive flow” can still be valid, as in reality the so-called constitutive secretion is finely regulated, as well detailed in this review. I am not suggesting a revision of the term, but just a second thought.
We thank this reviewer for this thought, which is actually something we also wondered about. We added a sentence in the summary section about that: "It is worth pointing out that the term "constitutive secretion" might need to be revisited, since this kind of secretion is also highly regulated, as we hope became clear from this review".
Minor points
Lanes 55-76; the statements are not supported by references.
We have added references to support the claims in those paragraphs.
Chapter 2.2 PKD Structure; lanes 144-154. The domain organization and the regulation of PKD are quite complex and not fully understood. I do not have a broad knowledge of the recent pertinent literature, but to my knowledge, even if the C1a domain is the major determinant for DAG binding and membrane recruitment, also the C1b domain can bind DAG, even if it appears to have a higher affinity for PMA. Finally, PKD1 and 2 contain a C-terminal PDZ domain responsible for interaction with several protein substrates, but this is not mentioned in the text. I also suggest evaluating the possibility of adding a schematic representation of the PKDs, as this protein family is a major regulator of carrier formation and is an important “transducer” of the signalling pathways described in this review.
We thank the reviewer for these suggestions, which have been included in the revised version of our manuscript. In addition, we have also added a new figure with the schematic representation of PKD domain structure (new Figure 1).
Lane 168, the correct definition of BARS is: Brefeldin-A ADP-ribosylation substrate
We have corrected that in our revised manuscript.
Reviewer 2 Report
The review entitled “Take me out: multiple signaling pathways for the biogenesis of TGN-to-plasma membrane transport carriers” submitted by Yuichi Wakana and Felix Campelo describes the trafficking pathway termed protein kinase D (PKD)-dependent pathway for constitutive secretion mediated by carriers of the TGN to the cell surface (CARTS), by highlighting the important role of organelle membrane contact sites and the regulation via lipid and protein (GPCR) signaling in this pathway.
This review is very interesting and the figures are of high quality. The Table 1 “Protein cargoes in TGN-to-PM transport carriers.” listing the described cargoes for each specific TGN to plasma membrane transport carriers will be of great use for researchers studying this pathway. The bibliography is very complete with the most recent articles being cited, even a reprint in bioRxiv treating the subject is included.
The figures are of high quality.
I have some minor comments:
- I suggest to add the MCSs abbreviation in the abstract to facilitate future retrieval of this excellent review by searches using MCSs as key word.
- Figure 2, it is not very clear from the figure that Sac1 mediates the catalytic PI4P to PI reaction.
- The legend of Figure 2 is very short; clearly the text is very clear and will allow the reader to perfectly understand, but indicating at least some of the used abbreviation (MCSs, SM, cer, chol) in the figure would be helpful for the non-expert reader.
Author Response
REVIEWER 2
The review entitled “Take me out: multiple signaling pathways for the biogenesis of TGN-to-plasma membrane transport carriers” submitted by Yuichi Wakana and Felix Campelo describes the trafficking pathway termed protein kinase D (PKD)-dependent pathway for constitutive secretion mediated by carriers of the TGN to the cell surface (CARTS), by highlighting the important role of organelle membrane contact sites and the regulation via lipid and protein (GPCR) signaling in this pathway.
This review is very interesting and the figures are of high quality. The Table 1 “Protein cargoes in TGN-to-PM transport carriers.” listing the described cargoes for each specific TGN to plasma membrane transport carriers will be of great use for researchers studying this pathway. The bibliography is very complete with the most recent articles being cited, even a reprint in bioRxiv treating the subject is included.
The figures are of high quality.
We thank the reviewer for the positive evaluation of our review.
I have some minor comments:
- I suggest to add the MCSs abbreviation in the abstract to facilitate future retrieval of this excellent review by searches using MCSs as key word.
We thank the reviewer for this suggestion. We have added the MCSs abbreviation in the abstract.
- Figure 2, it is not very clear from the figure that Sac1 mediates the catalytic PI4P to PI reaction.
We have adapted the figure to make this point more clear.
- The legend of Figure 2 is very short; clearly the text is very clear and will allow the reader to perfectly understand, but indicating at least some of the used abbreviation (MCSs, SM, cer, chol) in the figure would be helpful for the non-expert reader.
We have added the list of abbreviations used in that figure in the revised figure legend.
Reviewer 3 Report
Comments to the Authors
Overall, this review was well focused and written on a particular membrane traffic route from the trans-Golgi network (TGN) to the cell surface: PKD-dependent pathway for constitutive secretion mediated by CARTS.
However this reviewer still have several concerns.
1) Authors wrote in Abstract section, “In this review, we spotlight a particular membrane traffic route from the trans-Golgi network (TGN) to the cell surface: protein kinase D (PKD)-dependent pathway for constitutive secretion mediated by carriers of the TGN to the cell surface (CARTS)”.
However, the present title is simple “Take me out: multiple signaling pathways for the biogenesis of TGN-to-plasma membrane transport carriers”.
Since there are indeed multiple pathways and this review focus on special aspect of signaling pathways for the biogenesis of TGN-to-plasma membrane transport carriers, I feel better to change title of this review more focused on PDK-dependent pathway for constitutive secretion mediated by CARTS.
2) As authors mentioned in page 4, PKD is a family of serine/threonine-protein kinases that, in humans, encompasses 3 canonical members: PKD1, PKD2, and PKD3. Lack of each PKD showed different phenotypes in mouse.
However, in the text, only mentioned as PKD, thus readers do not know all three PKD act in same manners or not. Is it possible to say which isoform is responsible for particular aspect of signaling pathways for the biogenesis of TGN-to-plasma membrane transport carriers?
Author Response
REVIEWER 3
Overall, this review was well focused and written on a particular membrane traffic route from the trans-Golgi network (TGN) to the cell surface: PKD-dependent pathway for constitutive secretion mediated by CARTS.
We thank the reviewer for the positive evaluation of our review.
However this reviewer still have several concerns.
1) Authors wrote in Abstract section, “In this review, we spotlight a particular membrane traffic route from the trans-Golgi network (TGN) to the cell surface: protein kinase D (PKD)-dependent pathway for constitutive secretion mediated by carriers of the TGN to the cell surface (CARTS)”.
However, the present title is simple “Take me out: multiple signaling pathways for the biogenesis of TGN-to-plasma membrane transport carriers”.
Since there are indeed multiple pathways and this review focus on special aspect of signaling pathways for the biogenesis of TGN-to-plasma membrane transport carriers, I feel better to change title of this review more focused on PDK-dependent pathway for constitutive secretion mediated by CARTS.
We have changed the title to better describe the content of our review. The new title is: "The PKD-dependent biogenesis of TGN-to-plasma membrane transport carriers".
2) As authors mentioned in page 4, PKD is a family of serine/threonine-protein kinases that, in humans, encompasses 3 canonical members: PKD1, PKD2, and PKD3. Lack of each PKD showed different phenotypes in mouse.
However, in the text, only mentioned as PKD, thus readers do not know all three PKD act in same manners or not. Is it possible to say which isoform is responsible for particular aspect of signaling pathways for the biogenesis of TGN-to-plasma membrane transport carriers?
We thank the review for these suggestions. We have added a discussion on the different PKD knockout (and knock-in) mice (Section 2.1, last paragraph). We also added a more precise description of the 3 PKD proteins in Section 2.2. (last paragraph).